# CLR-Fact: Evaluating the Complex Logical Reasoning Capability of Large Language Models over Factual Knowledge

## Abstract

While large language models (LLMs) have demonstrated impressive capabilities across various natural language processing tasks by acquiring rich factual knowledge from their broad training data, their ability to synthesize and logically reason with this knowledge in complex ways remains underexplored. In this work, we present a systematic evaluation of state-of-the-art LLMs' complex logical reasoning abilities through a novel benchmark of automatically generated complex reasoning questions over general domain and biomedical knowledge graphs. Our extensive experiments, employing diverse in-context learning techniques, reveal that LLMs excel at reasoning over general world knowledge but face significant challenges with specialized domain-specific knowledge. We find that prompting with explicit Chain-of-Thought demonstrations can substantially improve LLM performance on complex logical reasoning tasks with diverse logical operations. Interestingly, our controlled evaluations uncover an asymmetry where LLMs display proficiency at set union operations, but struggle considerably with set intersections - a key building block of logical reasoning. To foster further work, we will publicly release our evaluation benchmark and code.

## 1 Introduction

Large language models (OpenAI, 2022; OpenAI et al., 2023) (LLMs) have shown impressive results in various natural language processing tasks by acquiring rich factual knowledge from diverse training corpora (Wei et al., 2022b;a; Ouyang et al., 2022). However, their ability to synthesize and utilize this knowledge for complex logical reasoning tasks involving operations like intersections, unions, and multi-hop reasoning remains largely unexplored (Bang et al., 2023; Huang et al., 2023).

While existing evaluations of factual knowledge primarily assess the memorization of simple facts (Thorne et al., 2018; Chen et al., 2023b; Sun et al., 2023; Huang et al., 2024), such as "What is the capital of France?" or "Which proteins are associated with lung cancer?", there is a lack of evaluation regarding how well language models can combine and synthesize those simple facts through multi-step logical reasoning. For example, while an LLM may know that Paris is the capital of France and that France borders Belgium, can it flexibly combine that knowledge to answer "What is the closest capital city to Paris besides the capital of France itself?" Many real-world applications require this type of complex reasoning over multiple facts, such as: (1) In healthcare, identifying patients that satisfy multiple criteria from electronic medical records for clinical trial recruitment. (2) In open-domain question answering, answering complex queries that build upon multiple pieces of provided information. Understanding the strengths and current limitations of LLMs in combining and logically reasoning over their broad factual knowledge can guide research toward developing more capable general reasoning systems.

To address this gap, we construct a new benchmark that utilizes high-quality knowledge graphs to automatically generate a diverse set of complex questions involving various reasoning patterns over factual knowledge. These questions require multi-step logical operations like intersections, unions, negations, and multi-hop reasoning over the knowledge graph entities and relations (Ren et al., 2020; Arakelyan et al., 2021; Bai et al., 2023b). We then systematically evaluate the performance of state-of-the-art large language models on this benchmark, leveraging different in-context learning techniques to probe their complex reasoning capabilities.

Additionally, we design controlled experiments focused specifically on evaluating the models' core capabilities for set operations like unions and intersections over entity sets, which form the building blocks of more complex logical reasoning.

Our findings suggest that large language models excel at reasoning over general knowledge but struggle with domain-specific knowledge like biomedical facts (§4.4). This indicates that while LMs can effectively leverage their broad training on widely available information sources like web pages and books, specialized knowledge domains pose greater challenges. We observed that LMs perform poorly on questions involving negations or set complementation (§4.4). This highlights a significant limitation in their ability to comprehend and reason with negative statements and set exclusion operations. In contrast, LMs exhibited proficiency at set union operations, but faced major difficulties with set intersections, suggesting an asymmetric grasp of set combinations versus identifying common elements across sets (§5.3). Meanwhile, our results verify the effectiveness of the Chain-of-Thought prompting technique for enhancing LM performance on complex questions requiring multi-step logical reasoning. By decomposing the reasoning process into explicit intermediate steps, this approach allows LMs to better handle the compositional reasoning demands of these intricate queries (§5.1). Additionally, we found that selecting demonstration examples based on semantic similarity to the query, such that the examples structurally align with the target reasoning pattern, provided an intuitive and effective method for improving LM performance through in-context learning (§5.1). Our key contributions are as follows:

- We propose CLR-Fact (Complex Logical Reasoning over Factual Knowledge), a novel evaluation framework that systematically assesses the capabilities of large language models to perform complex logical reasoning combining factual knowledge from knowledge graphs. The framework supports diverse reasoning patterns and domains through an ontology-driven approach.

- We construct a comprehensive evaluation benchmark consisting of 5,200 complex reasoning questions spanning 26 different logical patterns. The benchmark covers both general domain knowledge from a subset of Freebase as well as specialized biomedical domain knowledge extracted from PrimeKG.

- We conduct extensive experiments evaluating eight state-of-the-art large language models on the CLR-Fact benchmark, leveraging various in-context learning techniques. Additionally, we design focused evaluations probing the models' core capabilities on different set operations which form the basis for complex logical reasoning.

We believe the CLR-Fact framework and our detailed experimental analysis provide valuable insights into the strengths and limitations of current LLMs for complex logical reasoning over factual knowledge. To facilitate further research, we will publicly release the dataset and code upon acceptance.

## 2 Background and Related Work

### 2.1 Factuality Evaluation

Early approaches to evaluating factual consistency relied on n-gram based metrics (Papineni et al., 2002; Lin, 2004; Banerjee & Lavie, 2005), which assumed factual accuracy correlated with lexical overlap. More recent work has explored rich paradigms that combine entity analysis with question-answering and natural language inference. QAGS (Wang et al., 2020) extracts entities and generates questions to probe factual knowledge. Q2 (Honovich et al., 2021) frames factuality as a natural language inference task over entailment relations. Luo et al. (2023) generated diverse and well-coverage questions from knowledge graphs to evaluate factuality and robustness of language models. With the emergence of large language models, approaches like FActScore (Min et al., 2023) and FacTool (Chern et al., 2023) leverage the reasoning capabilities of LLMs to extract and verify facts against knowledge sources like Wikipedia. However, these prior methods primarily focus on assessing memorization of individual facts, failing to evaluate how LLMs synthesize and reason with factual knowledge in more complex ways involving multi-step inferences, logical operations, and reasoning over combinations of facts. Our work aims to fill this critical gap by constructing an evaluation framework specifically targeting LLMs' complex logical reasoning abilities over factual knowledge.

## 2.2 Complex Logical Reasoning over Knowledge Graphs

In another line of research, logical reasoning over knowledge graphs involves answering complex logical queries answering (Hamilton et al., 2019; Ren et al., 2020) aiming to use neural methods for answering complex logical queries derived from knowledge graphs. Various query encoding methods have been proposed to enhance the effectiveness and efficiency of logical reasoning, such as encoding queries with beta distribution (Ren & Leskovec, 2020), using neural link predictors for optimization search (Arakelyan et al., 2021), and employing sequential models to encode linearized queries (Bai et al., 2023b). In terms of the scope of logical formulas, Hamilton et al. (2019) initially introduced queries with relational projection and set intersection, which were later integrated with set union and set negation by Ren & Leskovec (2020). Moreover, Yin et al. (2023) transformed the scope from set operations to constraint satisfaction problems (CSPs). Most research in this area has focused on general-domain knowledge graphs, such as Freebase (Bollacker et al., 2008), YAGO (Suchanek et al., 2007), and NELL (Carlson et al., 2010). However, Bai et al. (2023a) extended the target to commonsense and eventuality knowledge graphs (Zhang et al., 2022). In this line of work, queries are represented in logical form but not in natural language form, making it difficult to directly apply complex reasoning over factual knowledge. This motivated us to construct a natural language-based complex reasoning benchmark for factual knowledge.

## 2.3 Reasoning with Large Language Models

In the evaluation process, we explored various methods to enhance the performance of language models in order to maximize their potential for conducting factual reasoning tasks. Here, we discuss some related work on LLM reasoning methods.

Among the in-context learning methods for reasoning with LLMs, the chain-of-thought (CoT) prompting method introduced by Wei et al. (2023) has emerged as a pivotal technique (Huang & Chang, 2023). CoT encourages explicit intermediate reasoning by incorporating ⟨input, chain of thought, output⟩ triples in the prompts. Subsequent iterations of this approach include Zero-shot-CoT by Kojima et al. (2022) and various applications across different domains such as code generation, multilingual tasks, and multimodal questions, demonstrating its versatility and effectiveness.

Complementing CoT, rationale engineering techniques such as rationale refinement, exploration, and verification have been developed to improve the quality and reliability of reasoning elicited from LLMs. These include complexity-based prompting (Fu et al., 2023) and algorithmic prompting (Zhou et al., 2022), as well as the use of LLMs themselves for rationale verification (Weng et al., 2023). Additionally, problem decomposition strategies have proven beneficial for tackling complex, compositional tasks by breaking them into smaller subproblems. Techniques like least-to-most prompting (Zhou et al., 2023), its dynamic variant decomposed prompting (Drozdov et al., 2022), and successive prompting (Dua et al., 2022) exemplify this approach and underscore the potential of CoT and related methods for advancing reasoning capabilities in LLMs. Meanwhile, Chen et al. (2023a) explored in-context methods to enhance logical reasoning of LLMs in the tasks of relation extraction and deductive reasoning.

# 3 CLR-Fact Evaluation Framework

The goal of this work is to comprehensively evaluate the complex logical reasoning capabilities of large language models when combining and reasoning over factual knowledge from both general domains and specific domains. While previous work has focused on assessing LLMs' memorization of simple facts, there has been less exploration into how well LLMs can synthesize and reason with those facts in complex ways involving logical operations like intersections, unions, negations, and multi-hop reasoning.

Formally, we aim to construct a benchmark covering a diverse set of complex logical reasoning questions over knowledge graphs. Given a knowledge graph $G$ containing factual triplets (*head*, *relation*, *tail*), the task is to generate natural language questions $q$ involving multi-step logical operations and constraints over the entities and relations in $G$. Then, for a given LLM $M$, we query $M$ with $q$ and evaluate whether $M$ can provide a set of correct answers $A$, which is the resulting entity set after applying the specified logical

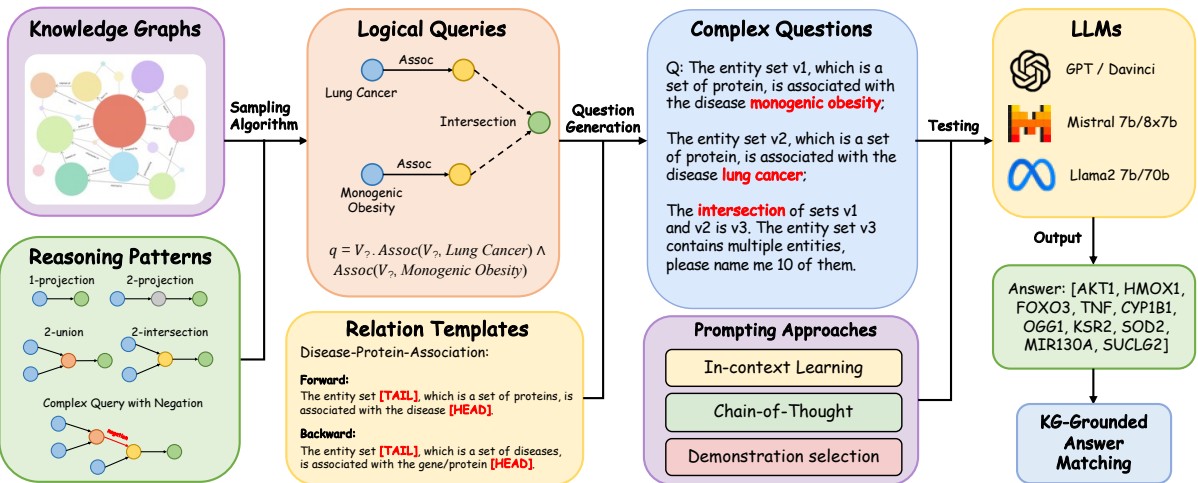

Figure 1: An overview of the **CLR-Fact** framework.

operations to the relevant subsets of $G$'s entities and relations mentioned in $q$. This process is demonstrated in Figure 1.

## 3.1 Logical Query Sampling on KG

In the first step, we conduct logical query answering over KG. We will introduce the definition of logical queries, and how to sample them. The complex queries on KG are defined in first-order logical form, and they can express complex semantics with the help of logical operators like *conjunction* $\wedge$, *disjunction* $\vee$, and *negation* $\neg$. We designed 26 types of complex queries (see Appendix J), and categorized them into four families based on their main logical operation: projection, intersection(conjunction), union(disjunction), and negation. The detailed definition of such queries can be found in the Appendix A. It's important to note that these logical queries are expressed using logical connectives, relations, and entities within the KG, not in natural language form. Therefore, we need to convert them into natural language sentences to make them compatible with LLMs.

## 3.2 Question Generation with Relation Templates

After acquiring logical queries, we proceed to transform them into natural language questions. First, we manually craft a natural language description, i.e. *Relation Template*, for each relation in two selected knowledge graphs. Each template encapsulates the precise semantics of the relationship, as well as the ontological categories of the potential head and tail entities. For instance, for a relationship denoting synergistic drug interactions, we create the template: "The entity set `[TAIL]`, which comprises drugs, exhibits a synergistic interaction with the drug `[HEAD]`." More examples are available in Appendix D. Using these templates, we can seamlessly translate each one-hop relational projection from the logical queries into natural language. Furthermore, we have developed a recursive tree-traversal algorithm (detailed in Appendix C) that connects and structures these one-hop relational projections with their corresponding logical operators, ultimately generating a coherent natural language question. Following the paradigm above, we construct our evaluation benchmark, consisting of 2,600 complex questions (26 reasoning patterns * 100 questions) for each knowledge graph. To demonstrate the reliability of our proposed question generation pipeline, we performed human evaluation on question quality of our benchmark via Amazon Mechanical Turk (AMT), with details in Appendix G.

## 4  Experiment

We conducted experiments on eight large language models over complex logical reasoning datasets generated from two knowledge graphs. In this section, we discuss our selection of knowledge graphs, evaluation metrics, selection of large language models, and the results of our main experiment.

### 4.1  Knowledge Graph Selection

Due to the diverse range of knowledge sources in the training corpus of large language models, including books, papers, web pages, and Wikipedia, it is not possible for a knowledge graph to fully encompass all this information. To ensure a comprehensive evaluation, the knowledge graph used in our benchmark construction should meet the following criteria: 1) The knowledge graph should contain high-quality, expert-curated factual knowledge from either a general or specific domain. 2) Entities should be represented in a machine-readable natural language format. 3) The knowledge graph should have comprehensive entity coverage to minimize false negatives under the open-world assumption. With those criteria, we selected two high-quality knowledge graphs for our dataset generation.

**FB15k-237**  FB15k-237 (Toutanova & Chen, 2015) is a commonly used knowledge graph in recent natural language processing research. It consists of high-quality triplets selected from the Freebase knowledge graph, covering general domain factual knowledge such as celebrities, films, organizations, locations, and awards, among others.

**PrimeKG**  PrimeKG (Chandak et al., 2023) is a large-scale biomedical knowledge graph that has been curated from 20 high-quality resources, biorepositories, and ontologies. It covers major pharmaceutical concepts and relationships, making it a reliable source for evaluating domain-specific factual knowledge in the biomedical field.

### 4.2  Evaluation Metrics

We obtain responses from LLMs to generated complex factual questions in the form of answer lists. Here, we introduce our method for evaluating the correctness of these answers.

**The Precision@10 Metric**  Since knowledge graphs may not include all entities that are factually correct for a complex question, false negatives are likely to occur if the traditional hit@K metric is used. To navigate this conundrum, we use the *precision@10* metric, which measures the precision of the first ten answers generated by LLMs. The metric is defined in Eq. (1), where $A_G$ denotes the answer set generated by the LLM, and $A_K$ denotes the answer set verified in the knowledge graph.

$$Precision@10 = \frac{|\{r \in A_G \mid r \in A_K\}|}{10} \tag{1}$$

**Answer Matching**  Exact-matching methods can be problematic in our benchmark as machine-generated answers may vary in format and structure. For instance, there may be long medical terms in the knowledge graph, and the LLM may generate the correct term but with different capitalization or hyphens, causing an exact match to fail. To address this issue, we employ the Jaro-Winkler text similarity (Winkler, 1990) for answer validation. We compare the evaluation results of different thresholds with results from human verification (with details in Appendix F), and set the final thresholds for FB15k-237 and PrimeKG as 0.90 and 0.97, respectively.

$$Jaro = \frac{1}{3}\left(\frac{m}{|s_1|} + \frac{m}{|s_2|} + \frac{m-t}{m}\right) \tag{2}$$

$$Jaro\_Winkler = Jaro + (l \cdot p \cdot (1 - Jaro)) \tag{3}$$

In the above formula, $m$ denotes matching characters, $t$ denotes transpositions, $s_1$ and $s_2$ are compared strings, $l$ denotes common prefix length, and $p$ denotes prefix scaling factor.

### 4.3  Models

We select several public accessible large language models for experiments. The selection of the large language models can be attributed to their state-of-the-art performance in various NLP benchmarks and their accessibility to the public. Below are brief technical details of each model:

**Llama-2** (Touvron et al., 2023) is an open-source foundation model built by Meta. It provides models ranging in size from 7B to 70B parameters, with a focus on improved safety and helpfulness. **Mistral-7b** (Mistral, 2023a) is one of the highest-performing open-source foundation models of the same parameter size. **Mixtral-8x7b** (Mistral, 2023b) is another open-source model by Mistral AI that leverages a mixture-of-experts (MoE) mechanism to improve performance. **Text-Davinci-003** (Ouyang et al., 2022) is a 175B question-answering language model built by OpenAI on top of GPT-3 (Brown et al., 2020). **GPT-3.5-turbo** (OpenAI, 2022) has been fine-tuned specifically for generating conversational text. It follows instructions in a prompt and provides human-like responses. **GPT-4** (OpenAI et al., 2023) is a new generation of GPT with scaled-up parameters and enhanced reasoning capabilities. **GPT-4o** (OpenAI, 2024) is the latest large language model released by OpenAI, achieving state-of-the-art performance in various NLP benchmarks across different modalities.

### 4.4  Main Result

Table 1 presents the experimental results of eight LLMs under 2-shot in-context learning. All models achieved substantially lower performance on PrimeKG compared to FB15k-237, indicating their limitations in reasoning over domain-specific knowledge (biomedical domain) versus general knowledge. Furthermore, the results for different reasoning patterns demonstrate that the models' performance significantly decreased when answering questions involving *Negation* operations, showing their poor ability to follow *set negation* instructions in complex logical reasoning. Additionally, the models' performance declined as the reasoning depth of the complex questions increased from 1 to 3, revealing that language models incur a performance cost when dealing with multi-hop questions that require deeper reasoning paths. Among all LLMs, GPT-4o achieved the best performance in both datasets, while other models such as GPT-4, Mixtral-8x7b, GPT-3.5-Turbo, and Text-Davinci-003 also performed considerably well.

## 5  Further Discussions

### 5.1  Study on Chain-of-Thought Prompting

Chain-of-Thought prompting (Wei et al., 2023) is an in-context learning technique that has proven effective in various reasoning tasks, including commonsense reasoning and arithmetic reasoning. This technique enhances LLMs' ability to solve complex reasoning questions by breaking down question-answer mappings into multiple intermediate reasoning steps. Table 2 illustrates the impact of Chain-of-Thought prompting in our benchmark study. We limited the number of demonstrations to four, recognizing that increased context length could detract from the model's performance, while too few demonstrations may not provide sufficient reasoning guidance for the model to capture. The results indicate that the Chain-of-Thought approach significantly improves reasoning performance across both datasets, particularly for questions involving negation operations.

Furthermore, Table 3 summarizes the improvements achieved using Chain-of-Thought across queries with different reasoning operation variety. Reasoning operation variety means the number of different logical operators in a logical query (detail in Appendix J). For instance, "4 types" indicates the presence of all four logical operations—projection, intersection, union, and negation—in the complex questions. The results suggest that Chain-of-Thought yields greater improvements in questions with a higher variety of operations, demonstrating its effectiveness in enhancing models' capabilities to handle complex reasoning patterns.

| Dataset | Model | Reasoning Pattern Family | | | | Reasoning Depth | | | Average |
|---|---|---|---|---|---|---|---|---|---|
| | | Pro. | Int. | Uni. | Neg. | 1-step | 2-steps | 3-steps | |
| FB15k-237 | Llama2-7b | 14.74 | 12.35 | 12.05 | 8.25 | 13.47 | 11.58 | 7.74 | 11.57 |
| | Llama2-70b | 28.12 | 21.92 | 22.55 | 14.84 | 22.09 | 21.67 | 19.01 | 21.32 |
| | Mistral-7b | 23.94 | 19.39 | 22.35 | 11.75 | 21.14 | 18.35 | 14.97 | 18.77 |
| | Mixtral-8x7b | 33.36 | 24.92 | 26.88 | 16.81 | 26.92 | 24.81 | 20.66 | 24.82 |
| | Text-Davinci-003 | 33.46 | **28.96** | 32.74 | 21.10 | 28.94 | _28.73_ | **26.88** | 28.45 |
| | GPT-3.5-turbo | 33.22 | 24.56 | 30.42 | 14.21 | 26.81 | 24.47 | 21.14 | 24.73 |
| | GPT-4 | _35.78_ | 27.02 | **39.87** | _22.14_ | **36.30** | 28.30 | 23.78 | _30.51_ |
| | GPT-4o | **38.44** | _28.39_ | _39.59_ | **25.00** | _35.71_ | **31.56** | _26.86_ | **32.25** |
| PrimeKG | Llama2-7b | 4.35 | 3.34 | 3.33 | 1.83 | 3.08 | 3.60 | 2.09 | 3.11 |
| | Llama2-70b | 7.96 | 6.71 | 5.63 | 3.19 | 5.25 | 6.37 | 4.97 | 5.67 |
| | Mistral-7b | 4.88 | 5.43 | 5.36 | 3.26 | 4.23 | 5.08 | 4.38 | 4.62 |
| | Mixtral-8x7b | 13.09 | 13.43 | 11.51 | 5.50 | 10.42 | 11.13 | 9.11 | 10.47 |
| | Text-Davinci-003 | 13.09 | 13.39 | 10.61 | 4.86 | 9.28 | 11.68 | 8.02 | 10.05 |
| | GPT-3.5-turbo | 13.97 | _16.51_ | 11.77 | 6.68 | 13.51 | 11.34 | 9.43 | 11.81 |
| | GPT-4 | _15.50_ | 14.44 | _18.35_ | _7.79_ | _14.54_ | _13.97_ | _10.60_ | _13.54_ |
| | GPT-4o | **18.63** | **22.56** | **20.15** | **12.55** | **19.86** | **17.90** | **14.56** | **18.02** |

Table 1: Main experiment result in precision@10 percentage. All models are tested under a 2-shot setting. Regarding "Reasoning Pattern Family", the pro. means the queries with the last operation of relational projection. The int./uni. means the queries with the last operations with logical intersection/union. Finally, the neg. means the queries with the last operation of set complement or negation. "Reasoning Depth" indicates the maximum number of consecutive relational projections in the complex question. The detailed query types, their reasoning pattern family and reasoning depths are presented in Appendix J.

| Dataset | Model | Reasoning Pattern Family | | | | Reasoning Depth | | | Average |
|---|---|---|---|---|---|---|---|---|---|
| | | Pro. | Int. | Uni. | Neg. | 1-step | 2-steps | 3-steps | |
| FB15k-237 | GPT-3.5-turbo | **32.60** | 22.84 | 31.61 | 10.64 | 25.13 | 21.95 | 22.93 | 23.36 |
| | +CoT | 32.29 | **24.43** | **33.62** | **24.33** | **30.25** | **27.96** | **25.31** | **28.33** |
| | Text-Davinci-003 | **38.02** | 27.06 | **35.98** | 19.35 | 29.86 | 29.19 | **28.29** | 29.27 |
| | +CoT | 37.90 | **27.20** | 34.44 | **25.38** | **31.98** | **31.55** | 26.69 | **30.78** |
| PrimeKG | GPT-3.5-turbo | 12.63 | 12.46 | 11.72 | 2.97 | 10.02 | 9.59 | 8.04 | 9.43 |
| | +CoT | **13.35** | **16.56** | **14.50** | **9.25** | **14.79** | **13.04** | **9.84** | **13.09** |
| | Text-Davinci-003 | **12.57** | **13.24** | **10.91** | 4.85 | 10.52 | **10.75** | 7.20 | **9.91** |
| | +CoT | 11.19 | 12.68 | 10.61 | **6.35** | **11.21** | 9.62 | **7.95** | **9.91** |

Table 2: The Precision@10 result with 4-shot Chain-of-Thought prompting.

## 5.2 Study on Demonstration Selection

Numerous studies have shown that the performance of LLMs is heavily dependent on the choice of in-context demonstrations (Lu et al., 2022; Zhao et al., 2021; Min et al., 2022). The term "Demonstration Selection" refers to the process of determining which examples are most beneficial for LLMs during in-context learning (Dong et al., 2023). Aiming for a flexible yet effective method, we utilized the text-embedding-ada-002 model to encode questions from an expanded dataset containing 1,000 complex questions from each of the six basic reasoning patterns. During the evaluation phase, we encode the input question using the same model and select the question with the highest cosine similarity score as our demonstration. According to the experimental result in Table 4, demonstration selection results in an average performance improvement of 12-25% across two datasets.

| Dataset | Reasoning Operation Variety | | | |
|---|---|---|---|---|
| | 1 type | 2 types | 3 types | 4 types |
| FB15k-237 | -2.99 | +1.32 | +10.75 | +14.90 |
| PrimeKG | -4.17 | +1.72 | +5.24 | +10.84 |

Table 3: Improvement with Chain-of-Thought prompting in questions with different operation varieties on GPT-3.5-turbo in Precision@10 (%)

| Dataset | Model | Demo | Reasoning Pattern Family | | | | Reasoning Depth | | | Average |
|---|---|---|---|---|---|---|---|---|---|---|
| | | | Pro. | Int. | Uni. | Neg. | 1-step | 2-steps | 3-steps | |
| FB15k-237 | GPT-3.5-turbo | highest | **33.44** | **27.61** | **32.20** | **18.20** | **31.57** | **25.45** | **21.91** | **27.12** |
| | | random | 33.22 | 24.56 | 30.42 | 14.21 | 26.81 | 24.47 | 21.14 | 24.73 |
| | | lowest | 31.26 | 21.98 | 28.12 | 10.80 | 24.07 | 22.09 | 18.16 | 22.10 |
| | Text-Davinci-003 | highest | **38.87** | **31.74** | **38.18** | **24.30** | **36.77** | **31.19** | **27.26** | **32.58** |
| | | random | 33.46 | 28.96 | 32.74 | 21.10 | 28.94 | 28.73 | 26.88 | 28.45 |
| | | lowest | 33.84 | 27.88 | 29.99 | 20.31 | 27.36 | 28.76 | 24.56 | 27.41 |
| PrimeKG | GPT-3.5-turbo | highest | **15.19** | 14.54 | **14.25** | **7.91** | **14.41** | **11.90** | **10.43** | **12.58** |
| | | random | 13.97 | **16.51** | 11.77 | 6.68 | 13.51 | 11.34 | 9.43 | 11.81 |
| | | lowest | 12.72 | 12.70 | 10.99 | 4.58 | 11.15 | 9.92 | 6.93 | 9.81 |
| | Text-Davinci-003 | highest | **17.99** | **16.56** | **16.79** | **7.94** | **15.38** | **14.64** | **11.35** | **14.29** |
| | | random | 13.00 | 13.39 | 10.61 | 4.86 | 9.28 | 11.68 | 8.02 | 10.05 |
| | | lowest | 12.38 | 12.01 | 10.68 | 5.37 | 8.54 | 11.71 | 7.82 | 9.74 |

Table 4: Experiment result of demonstration selection under 2-shot settings. "Highest" means demonstration has embeddings with the highest similarity score, and "Lowest" means demonstrations with lowest similarity scores.

The efficacy of demonstration selection underscores the importance of aligning language models with external knowledge sources. With advancements in Retrieval-Augmented Generation (RAG) systems (Lewis et al., 2021) and Vector Databases, there is promising potential to tailor complex questions with domain knowledge and retrieve them for inclusion in the LLM's context at inference time.

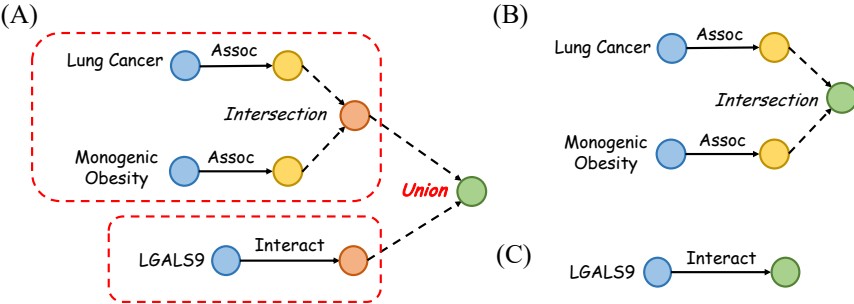

Figure 2: An illustration of set operation test. (A) refers to the reasoning pattern of the original complex question. (B) and (C) are the reasoning pattern of the two sub-questions of (A) before the final "union" operation.

## 5.3  Study on Set Operation Capabilities

The effective execution of logical operators, such as intersection and union over entity sets, is essential for models to perform multi-hop logical reasoning (Ren et al., 2020). Unlike many query answering models that perform set operations following fixed executions in the vector space, the set operations performed by LLMs

are opaque and implicit. Therefore, we propose evaluating set operations under a marginal setting. For each complex question ending with a set operation of *Intersection* or *Union*, we take the two sub-questions preceding the final set operation and test those sub-questions independently. Figure 2 provides an example of this experimental approach to set operations. We calculate a weighted average based on the ground truth answer set sizes for the two sub-questions and compare it with the score of the original question. Table 5 summarizes the experimental results of the set operations test for all four models. All models exhibit a performance decline when executing set operations in multi-hop reasoning, with the performance loss for *Intersection* being much greater than for *Union*. This outcome could be attributed to the set intersection's inherently more challenging nature, as it requires the exclusion of a number of answers present in the sub-questions from the final answer set, as opposed to the set union, where no answers from the sub-questions are excluded.

| Dataset | Model | Set Intersection | | | Set Union | | |
|---------|-------|--------|-------|------|--------|-------|------|
| | | Before | After | Drop | Before | After | Drop |
| FB15k-237 | GPT-3.5-turbo | 47.96 | 24.56 | 23.40 | 38.18 | 30.42 | 7.75 |
| | Text-Davinci-003 | 50.28 | 28.96 | 21.31 | 38.57 | 32.74 | 5.83 |
| | Mixtral-8x7b | 43.05 | 24.92 | 18.13 | 33.92 | 26.88 | 7.04 |
| PrimeKG | GPT-3.5-turbo | 57.39 | 16.51 | 40.88 | 26.40 | 11.77 | 14.63 |
| | Text-Davinci-003 | 49.82 | 13.39 | 36.43 | 21.51 | 10.61 | 10.90 |
| | Mixtral-8x7b | 43.38 | 13.43 | 29.97 | 21.42 | 11.51 | 9.91 |

Table 5: Experiment result of the set operation test. "Before" includes the weighted-average score on child questions before the final set operation, "After" includes the models' performance on the whole question. "Drop" indicates the performance drop due to executing the final set operation.

## 6 Conclusion

In summary, this work introduces CLR-Fact, a novel evaluation framework to systematically assess the complex logical reasoning capabilities of large language models over factual knowledge from knowledge graphs. Through extensive experiments, we find that while LLMs excel at reasoning over general world knowledge, they face significant challenges with specialized domains, negations, and core reasoning operations like set intersections. Techniques like Chain-of-Thought prompting can boost performance on complex multi-step reasoning tasks. Overall, our detailed analysis uncovers critical bottlenecks like handling negations and set intersections that should be addressed to develop more capable general reasoning systems.

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

## A  Definition of Logical Queries on KG

Answering logical is a process that involves asking questions about a knowledge graph $\mathcal{G} = (\mathcal{V}, \mathcal{R})$. $\mathcal{V}$ is a set of vertices (entities), and $\mathcal{R}$ is a set of relations between these entities. To express relations in logical expressions, each relation $r$ is defined as a function with two arguments representing two entities $v$ and $v'$. The value of $r(v, v')$ is 1 if there is a relation between entities $v$ and $v'$.

Queries are defined in first-order logical (FOL) forms, using logical operations such as existential quantifiers $\exists$, conjunctions $\wedge$, disjunctions $\vee$, and negations $\neg$. A query includes anchor entities $V_a \in \mathcal{V}$, existential quantified variables $V_1, V_2, ... V_k \in \mathcal{V}$, and a target variable $V_? \in \mathcal{V}$. The goal of the query is to find answer entities $V_? \in \mathcal{V}$ that satisfy the logical expression in the query. A query can be converted to a disjunctive normal form, which is a disjunction of several conjunctive expressions.

$$q[V_?] = V_?.\exists V_1, ..., V_k : c_1 \vee c_2 \vee ... \vee c_n, \tag{4}$$

$$c_i = e_{i1} \wedge e_{i2} \wedge ... \wedge e_{im}. \tag{5}$$

Each conjunctive expression $c_i$ is a conjunction of literals $e_{ij}$, where $e_{ij}$ is an atomic or negation of an atomic expression in forms such as $r(v_a, V)$, $\neg r(v_a, V)$, $r(V, V')$, or $\neg r(V, V')$. Here, $v_a \in V_a$ is an anchor entity, and $V, V' \in \{V_1, V_2, ..., V_k, V_?\}$ are distinct variables satisfying $V \neq V'$.

When a query is an existential positive first-order (EPFO) query, it includes only conjunctions $\wedge$ and disjunctions $\vee$, and no negations $\neg$. When the query is a conjunctive query, it includes only conjunctions $\wedge$, and no disjunctions $\vee$ or negations $\neg$.

# B    Sampling Algorithm of Logical Queries

In this section, we introduce the algorithm used for sampling the complex queries from a given knowledge graph. The detailed algorithm is described in Algorithm 1. For a given knowledge Graph $G$ and a query type $t$, we start with a random node $v$ to reversely find a query that has answer $v$ with the corresponding structure $t$. Basically, this process is conducted in a recursion process. In this recursion, we first look at the last operation in this query. If the operation is *projection*, we randomly select one of its predecessors $u$ that holds the corresponding relation to $v$ as the answer of its sub-query. Then we call the recursion on node $u$ and the sub-query type of $t$ again. Similarly, for *intersection* and *union*, we will apply recursion on their sub-queries on the same node $v$. The recursion will stop when the current node contains an anchor entity.

---

**Algorithm 1** Ground Query Type

---

**Require:** $G$ is a knowledge graph.

    **function** GROUNDTYPE$(T, v)$

        $T$ is an arbitrary node of the computation graph.

        $v$ is an arbitrary knowledge graph vertex

        **if** $T.operation = p$ **then**

            $u \leftarrow$ SAMPLE$(\{u|(u, v)\text{is an edge in } G\})$

            $RelType \leftarrow$ type of $(u, v)$ in $G$

            $ProjectionType \leftarrow p$

            $SubQuery \leftarrow$ GROUNDTYPE$(T.child, u)$

            **return** $(ProjectionType, RelType, SubQuery)$

        **else if** $T.operation = i$ **then**

            $IntersectionResult \leftarrow (i)$

            **for** $child \in T.Children$ **do**

                $SubQuery \leftarrow$ GROUNDTYPE$(T.child, v)$

                $IntersectionResult$.PUSHBACK$(child, v)$

            **return** $IntersectionResult$

        **else if** $T.operation = u$ **then**

            $UnionResult \leftarrow (u)$

            **for** $child \in T.Children$ **do**

                **if** $UnionResult.length > 2$ **then**

                    $v \leftarrow$ SAMPLE$(G)$

                $SubQuery \leftarrow$ GROUNDTYPE$(T.child, v)$

                $UnionResult$.PUSHBACK$(child, v)$

            **return** $UnionResult$

        **else if** $T.operation = e$ **then**

            **return** $(e, T.value)$

---

## C   Algorithm for Question Generation

In this section, we introduce the algorithm used for converting logical queries to natural language questions. The detailed algorithm is described in Algorithm 2. In general, our algorithm performs a post-order recursion to traverse through the computational graph (query tree) of the complex query. We designed natural language templates for each logical operations: intersection (conjunction), union (disjunction), and negation, and apply those templates in our recursion process to connect the one-hop sentences of relational projection. We use *index* as an non-local variable to store the ids for each set in the question, to guarantee the coherence of coreference between sentences.

---

**Algorithm 2** Convert Logical Query to Natural Language Question

---

**Require:**
   **function** REL_TEMP($head, relation, tail$) apply natural language template for relation projection in triplet ($head, relation, tail$)

**Require:**
   **function** LOG_TEMP($op, first, second$) apply natural language template for logical operation between sets $first$ and $second$

**Require:**
   **function** CAL_RIGHT($query\_tree$) calculate the size of right child of current $query\_tree$

**Require:** $index$ is a non-local variable.
   **function** QUERY2TEXT($query\_tree$)
      **if** $query\_tree.left\_child$ exists **then**
         $left \leftarrow$ QUERY2TEXT($query\_tree.left\_child$)

      **if** $query\_tree.right\_child$ exists **then**
         $right \leftarrow$ QUERY2TEXT($query\_tree.right\_child$)

      $index \leftarrow index + 1$
      $op \leftarrow query\_tree.operator$
      **if** $op = projection$ **then**
         $relation \leftarrow query\_tree.relation$
         **if** $query\_tree.left\_child$ is entity **then**
            $head \leftarrow query\_tree.left\_child$
         **else**
            $head \leftarrow index$ - 1
         $tail \leftarrow index$
         $root \leftarrow$ REL_TEMP($head, relation, tail$)
      **else if** $op = negation$ **then**
         $first \leftarrow index - 1$
         $second \leftarrow index$
         $root \leftarrow$ LOG_TEMP($negation, first, second$)
      **else**
         $first \leftarrow index -$ CAL_RIGHT($query\_tree$) $- 1$
         $second \leftarrow index$
         **if** $op = union$ **then**
            $root \leftarrow$ LOG_TEMP($union, first, second$)
         **else if** $op = intersection$ **then**
            $root \leftarrow$ LOG_TEMP($intersection, first, second$)
      **return** $left + right + root$

---

# D   Relation Templates

| Dataset | Relation Template Examples |
|---|---|
| FB15k-237 | The entity set [TAIL], which is a set of sports teams, is the team of professional athlete [HEAD]. |
| | The entity set [TAIL], which is a set of politicians, has worked in government positions in the country [HEAD]. |
| | The entity set [TAIL], which is a set of celebrities, has dated with the celebrity [HEAD]. |
| PrimeKG | The entity set [TAIL], which is a set of drugs, will cause the side effect of phenotype [HEAD]. |
| | The entity set [TAIL], which is a set of genes, is associated with the disease [HEAD]. |
| | The entity set [TAIL], which is a set of anatomical parts, is the place where the gene [HEAD] is expressed. |

Table 6: Examples of relation template in two datasets.

# E   Dataset Statistics

| Statistics | Reasoning Pattern Family | | | | Reasoning Depth | | | Average |
|---|---|---|---|---|---|---|---|---|
| | Pro. | Int. | Uni. | Neg. | 1-step | 2-steps | 3-steps | |
| Word Counts in Question | 71.39 | 95.03 | 97.78 | 105.32 | 77.99 | 89.51 | 132.65 | 93.37 |
| Num of Set Operations | 3.17 | 4.83 | 4.83 | 5.88 | 4.30 | 4.45 | 6.40 | 4.77 |

Table 7: Statistics on word counts and number of set operations in our CLR-Fact dataset.

# F   Threshold Selection for Answer Matching

Regarding the selection of answer matching threshold, we adopted human verification to determine the optimal threshold for precise evaluation of LLM outputs (GPT-3.5-turbo). We observed that in the biomedical domain, two distinct entities might bear similar names (e.g., gene/protein codes), thus a higher threshold compared to the general domain is required.

| Dataset (n=130) | Thresholds (Jaro-Winkler Similarity) | | | | | Human Verification |
|---|---|---|---|---|---|---|
| | 0.875 | 0.9 | 0.925 | 0.95 | 0.975 | |
| FB15K-237 | 27.08 | **23.92** | 21.51 | 19.72 | - | 23.64 |
| PrimeKG | - | 19.97 | 14.25 | 11.81 | **11.20** | 11.40 |

Table 8: Evaluation results of different thresholds compared with human verification in two datasets.

## G   Human Evaluation of Benchmark Quality

We performed a human evaluation of the quality of questions in our benchmark via Amazon Mechanical Turk (AMT). We randomly sampled 260 complex questions from two datasets, with each question evaluated by three human annotators. As shown in Table 9, over 98% of the questions were evaluated as *Valid* based on the majority vote. The standard for validity was defined as: "This question is very clear. One will be able to answer it given enough factual information."

| Dataset | Validity by Individual (%) | Validity by Majority Vote (%) |
|---------|:--------------------------:|:-----------------------------:|
| FB15K-237 | 94.87 | 98.46 |
| PrimeKG | 92.31 | 98.46 |

Table 9: Results of human evaluation on question quality from two datasets.

## H   Case Study

Table 10 provides an example of how errors may occur during the complex reasoning process in LLMs. We observed that LLM may hallucinate when performing one-hop relational projections, and likely to confuse set intersections with set unions.

| **Example** | |
|---|---|
| Question | The entity set v1, which is a set of actors, has performed in the movie The Expendables. The entity set v2, which is a set of movies, is produced by Joel Silver. The entity set v3, which is a set of actors, has performed in the movie in the entity set v2. The intersection of sets v1 and v3 is v4. Please name 10 entities from v4. |
| Answer | Firstly, the entity set v1 is the actors who have performed in the movie The Expendables. Then, the entity set v2 is the movies that are produced by Joel Silver, which could include The Expendables. **(hallucination)** Then, the entity set v3 is the actors who have performed in the movie in the entity set v2, which means that the actors in v3 should be the same as the actors in v1. Finally, the entity set v4 should be the intersection of v1 and v3, which means that the actors in v4 should be the same as the actors in v1 and v3. **(error in reasoning with set intersection)** So, my answer to this question is: Sylvester Stallone, Jason Statham, Jet Li,... **(and seven incorrect answers due to hallucination and failing to exclude during set intersection)** |

Table 10: An example QA performed by Mixtral-8x7b model on question type (i,(p,(e)),(p,(p,(e)))). The model are prompted with CoT demonstrations to illustrate intermediate reasoning steps.

# I  Distribution of Demonstration Embeddings

Figure 3 visualizes the distribution of vector embeddings of demonstrations using t-SNE (van der Maaten & Hinton, 2008) for dimensionality reduction. It can be observed that the proximity of vector embeddings correlates strongly with the ontological categories of the answers; that is, embeddings of similar answer categories tend to cluster together. By selecting demonstrations with similar answer categories, we subtly align the language model with specific domain knowledge within the context. Furthermore, the vector embeddings appear to be implicitly related to the reasoning pattern of the questions, thus aiding in the selection of questions that share similar reasoning processes.

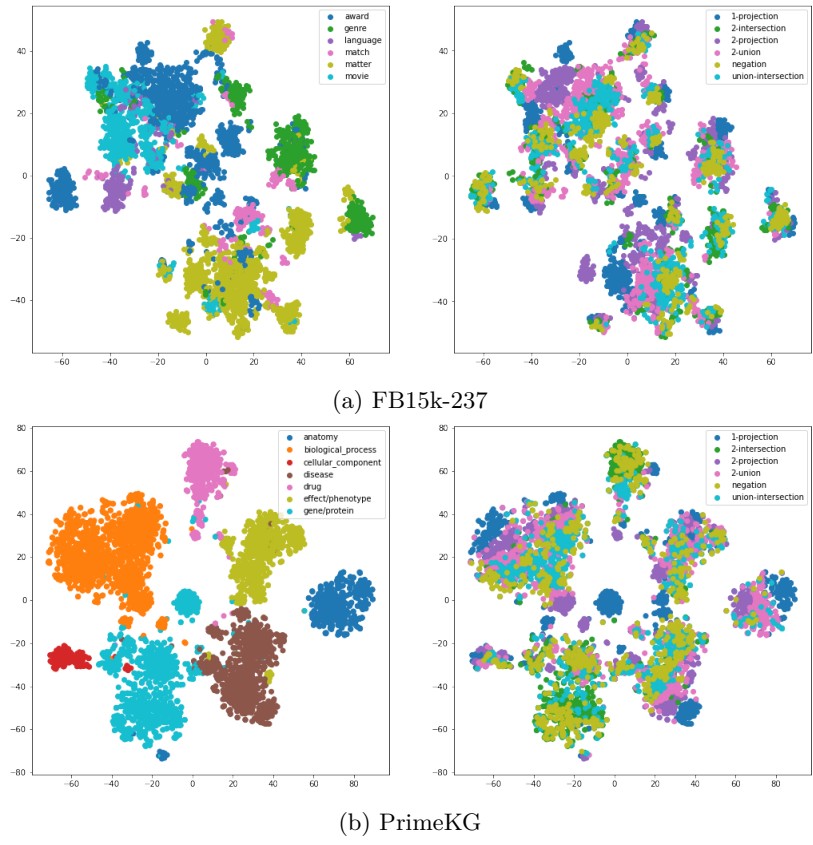

(a) FB15k-237

(b) PrimeKG

Figure 3: T-SNE visualization of sentence embeddings of demonstrations. For both two datasets, the left ones are categorized on the ontological type of the corresponding answer sets, while the right ones are categorized on reasoning pattern of the questions.[1]

---

[1]For clearness, we only included 6 major ontology types in our visualization for FB15K-237.

## J    Logical Query Types

The logical formula, reasoning depth, and operation variety of each query types in our dataset are presented in Table 11.

| Family | Lisp-like Formula of Query Type | Reasoning Depth | Operation Variety |
|---|---|---|---|
| Projection | (p,(e)) | 1 | 1 |
| | (p,(p,(e))) | 2 | 1 |
| | (p,(p,(p,(e)))) | 3 | 1 |
| | (p,(i,(p,(e)),(p,(e)))) | 2 | 2 |
| | (p,(i,(n,(p,(e))),(p,(e)))) | 2 | 3 |
| | (p,(u,(p,(e)),(p,(e)))) | 2 | 2 |
| Intersection | (i,(p,(e)),(p,(e))) | 1 | 2 |
| | (i,(p,(e)),(p,(p,(e)))) | 2 | 2 |
| | (i,(p,(p,(e))),(p,(p,(e)))) | 2 | 2 |
| | (i,(p,(p,(p,(e)))),(p,(p,(p,(e))))) | 3 | 2 |
| | (i,(i,(p,(e)),(p,(e))),(p,(e))) | 1 | 2 |
| | (i,(u,(p,(e)),(p,(e))),(p,(e))) | 1 | 3 |
| Union | (u,(p,(e)),(p,(e))) | 1 | 2 |
| | (u,(p,(e)),(p,(p,(e)))) | 2 | 2 |
| | (u,(p,(p,(e))),(p,(p,(e)))) | 2 | 2 |
| | (u,(p,(p,(p,(e)))),(p,(p,(p,(e))))) | 3 | 2 |
| | (u,(i,(p,(e)),(p,(e))),(p,(e))) | 1 | 3 |
| | (u,(u,(p,(e)),(p,(e))),(p,(e))) | 1 | 2 |
| Negation | (i,(n,(p,(e))),(p,(e))) | 1 | 3 |
| | (i,(n,(p,(e))),(p,(p,(e)))) | 2 | 3 |
| | (i,(n,(p,(p,(e)))),(p,(e))) | 2 | 3 |
| | (i,(n,(p,(p,(e)))),(p,(p,(e)))) | 2 | 3 |
| | (i,(n,(p,(p,(e)))),(p,(p,(p,(e))))) | 3 | 3 |
| | (i,(n,(p,(p,(p,(e))))),(p,(p,(p,(e))))) | 3 | 3 |
| | (i,(n,(i,(p,(e)),(p,(e)))),(p,(e))) | 1 | 3 |
| | (i,(n,(u,(p,(e)),(p,(e)))),(p,(e))) | 1 | 4 |

Table 11: All 26 query types with their corresponding reasoning depth and operation variety in our benchmark. We applied the lisp-like formula (Wang et al., 2021) to represent the structure of logical queries.

# K    Results on All 26 Reasoning Patterns

Experimental results of three LLMs on all 26 reasoning patterns are presented in Table 12. We observe that LLMs generally perform worse when the number of operators and reasoning depth increases. One exception is that the performance on the pattern `(p,(p,(p,(e))))` is slightly higher than on `(p,(p,(e)))`. This could be explained by the query sampling mechanism, as queries with longer reasoning chains are more likely to be sampled from head knowledge (Sun et al., 2024) instead of tail knowledge, thus implicitly resulting in higher scores in LLM evaluation.

| Reasoning Pattern | Llama2-70b | Mixtral-8x7b | GPT-4 | Average |
|---|---|---|---|---|
| (p,(e)) | 32.91 | 45.76 | 46.20 | 41.62 |
| (p,(p,(e))) | 30.86 | 34.90 | 34.80 | 33.52 |
| (p,(p,(p,(e)))) | 33.36 | 37.96 | 39.18 | 36.84 |
| (p,(i,(p,(e)),(p,(e)))) | 23.37 | 28.84 | 34.25 | 28.82 |
| (p,(i,(n,(p,(e))),(p,(e)))) | 22.08 | 23.72 | 26.88 | 24.22 |
| (p,(u,(p,(e)),(p,(e)))) | 26.16 | 28.98 | 33.38 | 29.51 |
| (i,(p,(e)),(p,(e))) | 27.89 | 35.05 | 37.35 | 33.43 |
| (i,(p,(e)),(p,(p,(e)))) | 22.09 | 30.06 | 24.95 | 25.70 |
| (i,(p,(p,(e))),(p,(p,(e)))) | 18.28 | 23.45 | 23.33 | 21.69 |
| (i,(p,(p,(p,(e)))),(p,(p,(p,(e))))) | 19.03 | 17.04 | 25.83 | 20.63 |
| (i,(i,(p,(e)),(p,(e))),(p,(e))) | 26.90 | 27.42 | 31.29 | 28.54 |
| (i,(u,(p,(e)),(p,(e))),(p,(e))) | 17.34 | 16.51 | 19.39 | 17.74 |
| (u,(p,(e)),(p,(e))) | 25.13 | 34.33 | 47.90 | 35.78 |
| (u,(p,(e)),(p,(p,(e)))) | 23.05 | 30.09 | 40.40 | 31.18 |
| (u,(p,(p,(e))),(p,(p,(e)))) | 23.76 | 21.04 | 33.26 | 26.02 |
| (u,(p,(p,(p,(e)))),(p,(p,(p,(e))))) | 23.47 | 22.10 | 34.40 | 26.66 |
| (u,(i,(p,(e)),(p,(e))),(p,(e))) | 17.11 | 21.14 | 35.65 | 24.63 |
| (u,(u,(p,(e)),(p,(e))),(p,(e))) | 22.80 | 32.55 | 47.60 | 34.32 |
| (i,(n,(p,(e))),(p,(e))) | 14.91 | 19.63 | 29.35 | 21.30 |
| (i,(n,(p,(e))),(p,(p,(e)))) | 23.44 | 20.18 | 22.25 | 21.96 |
| (i,(n,(p,(p,(e)))),(p,(e))) | 12.98 | 17.71 | 22.58 | 17.76 |
| (i,(n,(p,(p,(e)))),(p,(p,(e)))) | 12.27 | 13.93 | 15.22 | 13.81 |
| (i,(n,(p,(p,(e)))),(p,(p,(p,(e))))) | 8.43 | 13.53 | 11.00 | 10.98 |
| (i,(n,(p,(p,(p,(e))))),(p,(p,(p,(e))))) | 10.77 | 12.67 | 8.50 | 10.64 |
| (i,(n,(i,(p,(e)),(p,(e)))),(p,(e))) | 22.30 | 22.32 | 34.80 | 26.48 |
| (i,(n,(u,(p,(e)),(p,(e)))),(p,(e))) | 13.65 | 14.53 | 33.43 | 20.54 |
| Average | 21.32 | 24.82 | 30.51 | 25.55 |

Table 12: Results of three LLMs on each reasoning pattern in the FB15K-237 dataset under 2-shot settings.

