# OpenReview forum: "CLR-Fact: Evaluating the Complex Logical Reasoning Capability of Large Language Models over Factual Knowledge"
_TMLR — Withdrawn by Authors_

### Review · Reviewer_Jr1q · 2024-08-12

**Summary Of Contributions:**

The paper creates a new evaluation for measuring logical reasoning capabilities of LLMs, named CLR-Fact. The evaluation set is created with factual knowledge base with known correct relationship between entities. It turns those relations into natural language questions about logical set operations (union, intersection, negation etc), and ask LLMs to give the result set. The correctness is measured by precision@10 of the returned set.

**Audience:**

Yes

**Claims And Evidence:**

No

**Requested Changes:**

- Show full prompt to the model, including how the source set content are provided.
- Explain why JW algorithm was chosen for answer matching. What aspect of it makes it a good fit?
- Choosing 0.90 and 0.97 as the threshold seems brittle and may overfit to the setup. Can we have a better choice or show that this can be safely generalized?
- Explain why we use N=10 as the size for precision@N.
- (optional) maybe we can have Google's Gemini-1.5-Pro and Anthropic's Claude Sonnet 3.5 model in the experiment too as it's showing strong performance on recent leaderboards.

**Strengths And Weaknesses:**

Strength
- Reasoning is a key and difficult capability of LLMs. As current models compete fiercely (and probably overfitting) existing eval sets, a new evaluation is always welcomed.
- The construction of the evaluation guaranteed the correctness of the result.
- The evaluation can be run automatically, with no expensive human raters.

Weakness
- Most importantly, how do we ensure the answer is not in any LLM's training data, esp when it's more about world knowledge? e.g. the model may have seen and remember result of some logical questions. The paper shows that domain specific questions is harder than common knowledge ones, could this be a sign of that?
- Set operation is just part of complex logical reasoning. If the evaluation won't cover other cases, it's better to narrow the scope of the claim.
- The answer matching algorithm seems brittle. Have we considered using LLM as judges to increase flexibility? have we considered using deterministic output (choices, labels) to make the answer matching easier?
- I didn't see any description on how do we pass the sets to the model. It's surely not just by mentioning "v1". The format of the actual full prompt could have a significant effect on the outcome. It's better to provide that.

---

### Review · Reviewer_kmFQ · 2024-08-16

**Summary Of Contributions:**

This paper presents a systematic evaluation of  LLMs’ complex logical reasoning abilities, where authors use a  benchmark of automatically generated complex reasoning questions for evaluation.
Authors discover that LLMs excel at reasoning over general world knowledge but face significant challenges with specialized domain-specific knowledge.

**Audience:**

Yes

**Broader Impact Concerns:**

No.

**Claims And Evidence:**

No

**Requested Changes:**

1.  The caption of Fig. 1 does not contain any useful information or explain Fig.1, which makes Fig. 1 hard to understand. Authors does not explain the relation and the usage of each module of Fig.1.

2. This paper is not self-contained. Authors define different types of queries and such definitions are used hereafter, but such definitions are presented in Appendix. This layout makes this paper hard to understand from the beginning.

Besides, too many technical details are reported in different Appendix. For example, surprisingly, technical details of section 3.2 are totally introduced in three Appendix, rather not in the main paper.

3. Since authors in Section 3 do not introduced the proposed framework clearly (For example, what is a recursive tree-traversal algorithm?), the novelty of this paper is in doubt, because i cannot know the exact function of each module of the proposed framework.

4. How to generate a benchmark? Authors have clarified that one of their contribution is "construct a comprehensive evaluation benchmark consisting of 5,200 complex reasoning questions spanning 26 different logical patterns." However, authors do not introduce or explain how to construct such a benchmark, and do not show any example. Thus, the convincingness of this paper is in doubt.

5. Details of each experiment is missing. However, authors only introduce the result of each experiment, without introducing how each experiment is performed. Thus, the convincingness of experimental results and conclusions are limited.

**Strengths And Weaknesses:**

This paper does not even introduce clearly the most important part --- the proposed framework, where too many technique details are missing.
More crucially, the novelty and convincingness of this paper is in doubt. For example, authors do not introduce how to realize on of their claimed contribution --- constructing a benchmark.

---

### Review · Reviewer_N6vn · 2024-08-20

**Summary Of Contributions:**

The authors propose to study the capacity of LLMs to perform a more advanced form of reasoning than simple information retrieval. To do so they propose to use Knowledge Graphs (KG) combined with FOL  to formulate a set of 26 types of complex queries (with associated ground truth) including projection, intersection(conjunction), union(disjunction), and negation and use a template mechanism to convert the logical formulation in plain language. They analyse a variety of LLMs of different sizes and offer some insights on the reasoning capacity of LLMs.

**Audience:**

Yes

**Claims And Evidence:**

Yes

**Requested Changes:**

1. If one wants to (additionally?) evaluate the capacity of LLMs to perform logical operations independently of their factual knowledge, why not providing all information regarding the set members in the prompt? One could then study the dependence of the performance w.r.t. the size of these sets.
2. The author have developed a template that encapsulates the precise semantics of the relationship. How robust are the results to changes of these templates?
3. Addressing the questions reported in the Weaknesses section.

**Strengths And Weaknesses:**

Strengths:
1. The notion of probing and evaluating the reasoning capacity of LLMs is of interest.
2. The authors develop a clear template that uses a KG and a technique to sample queries and finally transform them into plain language.

Weaknesses:
1. The contribution to the performance results of LLMs' lack or reasoning and lack of factual knowledge is unclear: the experiment in table 5 seems to address the issue by analysing a specific logical step (union or intersection) given the quality of the answers to the sub-questions that have to be combined via union or intersection. But how is this taken into account in the other cases? I.e. how do we know that the LLM incorrect answer is because it did not know all the elements in the required sets or because it could not perform the logical operation.
2. Certain results are reported, but the quality of the paper would be enhanced if some intuitions could be offered as to why such results happen. For example, why is the LLM answer quality depending on the domain? Is it because specialised knowledge is more intrinsically difficult? or simply because it refers to entities that are more infrequent? What experiments could be devised to ascertain which is the reason?
The same with the finding that performance significantly decreased when answering questions involving negation operations. Why? is it because negations are harder to express using matrix operations? or is it because there are not many examples in the training corpus that use negations? How could we quantify this? Can we train a LLM on a balanced dataset w.r.t. negations and observe a change in the capacity of handling negations?
Also regarding the finding that performance declines as the reasoning depth of the complex questions increased from 1 to 3, why? is it because of an increase in complexity? or again is it because there are not many examples in the training corpus that use larger depths of reasoning? How could we quantify this?
3. The results suggest that Chain-of-Thought yields greater improvements in questions with a higher variety of operations. Does this mean that LLM are just performing in context learning? i.e. one needs to show the type of reasoning in the prompt for the LLM to use it in the answer? How can we validate such hypothesis?

---

### Note · Authors · 2024-09-03

I have read and agree with the venue's withdrawal policy on behalf of myself and my co-authors.